# Variations in Channel Centerline Migration Rate and Intensity of a Braided Reach in the Lower Yellow River

**Junqiang Xia** [1,*]**, Yingzhen Wang** [1]**, Meirong Zhou** [1]**, Shanshan Deng** [1]**, Zhiwei Li** [1] **and Zenghui Wang** [1,2]

[1] State Key Laboratory of Water Resources and Hydropower Engineering Science, Wuhan University, Wuhan 430072, China; wang_yz@whu.edu.cn (Y.W.); zmr@whu.edu.cn (M.Z.); dengss@whu.edu.cn (S.D.); lizw2003@whu.edu.cn (Z.L.); wang_zh@whu.edu.cn (Z.W.)

[2] College of Water Resources and Architectural Engineering, Northwest A & F University, Yangling 712100, China

**\*** Correspondence: xiajq@whu.edu.cn

**Abstract:** The Yellow River (YR) covers three climatic zones including arid region, semi-arid region and temperate monsoon region, with frequent appearance of flow intermittence in the Lower Yellow River (LYR) before 1999. Channel migration occurs frequently in braided rivers, which is a major focus of study in geomorphology and river dynamics. The braided reach in the LYR is featured by a complexly spatio-temporal variation in channel migration parameters owing to the varying condition of flow and sediment. It is crucial to investigate the migration characteristics of channel centerline for the sake of fully understanding channel evolution. A detailed calculation procedure is proposed to quantify migration rates and intensities of channel centerline at section- and reach-scales, using the measurements of remote sensing images and cross-sectional topography. Migration rates and intensities of channel centerline at section- and reach-scales from 1986 to 2016 were calculated, with the characteristics and key factors to control the migration intensity of channel centerline being identified quantitatively. Calculated results indicate that: (i) the mean probability of centerline migrating toward the left side was approximately equal to the probability of rightward migration from a long-term sequence; (ii) the mean reach-scale migration rate of channel centerline was reduced from 410 m/yr in 1986–1999 to 185 m/yr in 1999–2016, with a reduction of 55% owing to the Xiaolangdi Reservoir operation in 1999, and the mean reach-scale migration intensity of channel centerline was decreased from 0.28 to 0.16 m/(yr·m), with a reduction of 43%; (iii) the incoming flow-sediment regime was a dominant factor affecting the degree of channel migration, although the channel boundary conditions could influence the intensity of channel migration; and (iv) the reach-scale migration intensity of channel centerline can be written as a power function of the previous two-year average incoming sediment coefficient or fluvial erosion intensity, and the reach-scale migration intensities of channel centerline calculated using the proposed relations are generally in close agreement with the measurements over the period of 30 years.

**Keywords:** channel centerline; migration rate; migration intensity; braided reach; Lower Yellow River

## 1. Introduction

Braided channel pattern is one of the common river patterns of natural rivers, which is widely distributed throughout the world. The Mengjin–Gaocun reach of the Lower Yellow River (LYR) in China and the Jamuna River in Bangladesh belong to typical braided rivers [1,2]. These rivers have a number of central bars or islands, which lead to multiple channels at low flows, and such a planform geometry usually results in frequent lateral channel shifting. For example, the main-channel near the Liuyuankou section of the braided reach in the LYR shifted to and fro along the north-south direction during the 1954 flood season, with the migration amplitude greater than 6 km within 24 h [3]. The Jamuna River is quite active, and the channel shifts frequently, with lateral migration rates greater than 500 m/yr [4]. These fluvial processes of braided rivers can cause unfavorable effects

such as inducing the collapse of floodplain banks and threatening the safety of levees. Therefore, channel migration processes in braided rivers have aroused major concern in the study of river dynamics and geomorphology [5–7].

However, previous studies on channel evolution of braided rivers usually focused on investigations into the variation in incoming flow-sediment regime, channel aggradation and degradation, and adjustments in cross-sectional geometry [8–12], and seldom conducted a quantitative analysis of channel migration in braided rivers [13]. Channel migration plays a vital role in morphological changes of braided rivers, and usually includes various lateral shiftings of bankline, thalweg, main streamline and channel centerline [2,5,14]. A number of data sources derived from historical maps, cross-sectional topography and remote sensing images can be used to determine these changes [10,11,13,15–17].

Leys and Werritty [18] reported that an image processing software (ER Mapper) was applied to handle aerial photographs and scanned maps, in order to conduct a detailed analysis of planform changes of historical river channels, and the lateral shifting of channel centerline was estimated in the selected river reach of the Cleekhimin Burn. Richard et al. [16] adopted a geographic information system (GIS) from digitized aerial photographs of the active river channel without vegetation, and quantified the reach-scale lateral migration in the Cochiti reach of the Rio Grande over the period 1918–2001. Furthermore, many previous studies were conducted in the braided reach of the LYR. Wu et al. [19] examined the influence of artificial river regulation projects on lateral migration intensity of main-flow path, and it was found that the mean shifting rate of main-flow path was reduced from 425 m/yr in 1949–1960 to 160 m/yr in 1975–1990 owing to the implementation of river training works. Chen et al. [14] pointed out that lateral shifting of main streamline was dominant in channel planform adjustments of the braided reach, and an analysis of historical river regime maps indicated that the mean shifting rate of main streamline was reduced by 16–34% in response to the operation of the Xiaolangdi Reservoir (2000–2008), as compared with the value during the operation of the Sanmenxia Reservoir (1960–1964). A reach-averaged method proposed by Xia et al. [10] was used to determine the bankfull channel dimensions in the braided reach, with the reach-scale bankfull width increasing by about 400 m in 1999–2012, which approximately represented the magnitude of bankline retreat at reach-scale over the past 13 years. Furthermore, Li et al. [13] calculated the migration widths and intensities of channel thalweg at both section- and reach-scales in the braided reach in 1986–2015, and it was found that thalweg-migration width and intensity at reach-scale were reduced, respectively, by 47% and 35% after the initial operation of the reservoir in 1999. However, these traditional methods using a limited number of historical maps or cross-sectional topography are generally localized in extent, and hence cannot accurately investigate the detailed channel migration processes of alluvial rivers [20–22].

As a multi-temporal, multi-spectral, cost-effective and high-resolution information source, the technology of remote sensing has been widely used to investigate changes of channel morphology in alluvial rivers, with specified methodologies or software being developed to quantify channel properties [17,21,23–26]. Rowland et al. [16] presented the methodologies of a set of analysis algorithms to determine riverbank erosion and accretion, which can be used to analyze the planform variations of various river morphologies. Remote sensing technology can be used to quantify planform changes in alluvial rivers, covering channel centerline migration and riverbank shifting [7,20,21,27–30]. The spatio-temporal changes of riverbanks and channel centerlines in the active Yellow River Delta were systematically quantified, using the technologies of satellite remote sensing and GIS [20]. Peixoto et al. [6] used Landsat TM imagery to investigate spatial and temporal migration of three reaches in western Brazilian Amazonia over a 21-year period, and found that the rates of annual lateral erosion and vegetated land accretion were well-balanced along these three reaches. Using eight dry-season satellite images of the Ganges River in Bangladesh, morphological changes in this reach were assessed, and the bankline shifting rates of the river were also quantified under four different periods in 1973–2009, with the results indicating that both the left and right riverbanks underwent significant migration

processes in response to varying rates of bank erosion and accretion [31]. Various satellite imageries were employed to quantify the planform migration of the Lower Jingjiang Reach of the Middle Yangtze River, and the mean channel centerline migration rate decreased from 31.1 m/yr in 1983–1988 to 11.6 m/yr in 2009–2013 [21]. Kong et al. [17] investigated planform changes in the LYR in 1987–2017 using multi-temporal remotely sensed images, and it was found that the mean migration rate of channel centerline in the braided reach varied around 30–100 m/yr after the Xiaolangdi Reservoir operation. Therefore, an integration of remote sensing and GIS can be used to quantify channel migration processes in alluvial rivers, covering the lateral shifting rates of channel centerline and bankline. However, these studies are seldom integrated with the results obtained from traditional approaches based on the discipline of river dynamics, and therefore they cannot present the quantitative relationships between channel morphometric parameters and incoming flow-sediment factors.

The regime of flow and sediment was altered dramatically in the braided reach owing to the operation of the Xiaolangdi Reservoir [8,10,11,14,32]. In consequence, significant channel evolution occurred, including continuous channel degradation and remarkable migration of channel centerline. Therefore, it is appropriate to determine the spatio-temporal variations in migration rate and intensity of channel centerline in the braided reach, for the sake of fully understanding the characteristics of fluvial processes. In this study, a fusion approach of data from multiple sources at various scales has been adopted to investigate the spatio-temporal variations in migration rate and intensity of channel centerline in the braided reach over the period from 1986 to 2016. These multiple-source data cover satellite images, cross-sectional topography, discharges and sediment concentrations at different hydrometric sections. The major aims of the current study are: (i) to develop an integrated procedure to determine the migration rate and intensity of channel centerline at reach-scale; (ii) to estimate the morphometric parameters at reach-scale to depict the channel morphology; and (iii) to analyze the key factors that influence the migration intensity of channel centerline and establish empirical relationships between migration intensity and different flow-sediment factors based on investigations of their varying characteristics.

## 2. Study Area

The 5464 km long Yellow River originates from the Bayankala Mountains and flows from west to east through nine administrative regions, and finally extends eastward to the Bohai Sea [33–35], which covers three climatic zones including arid region, semi-arid region and temperate monsoon region. Before emptying into the Bohai Sea in Shandong Province, the 786-km long Lower Yellow River flows from Mengjin (abbreviated to MJ) in Henan Province of central China to the northeast across the North China Plain, with a basin area of only 23,000 km² accounting for a mere 3% of the total area (Figure 1a). The total drop in elevation of the LYR is 93.6 m, with an average channel slope of 0.012%. The LYR receives water and sediment from the mainstream of the Middle Yellow River and two tributaries of the Yiluo River and the Qin River. Excessive sediment deposition in the LYR caused the riverbed level in Kaifeng, Henan Province to be 10 m higher than the level around the surrounding zone. There are three distinct reaches in the LYR according to their different geomorphological characteristics, including a braided reach from Mengjin to Gaocun, a transitional reach from Gaocun to Taochengpu and a meandering reach from Taochengpu to Lijin in proper sequence. There is a narrow main-channel zone with the width ranging between 0.5 and 4.5 km in the LYR, with the area of main-channel zone accounting for about 20% of the total channel area; however, the widths of two floodplains on both sides vary from 5 to 20 km. The braided reach is relatively straight, with the average curvature coefficient of 1.15 and the planform geometry characterised by alternated wide and narrow sub-reaches, with rapid morphological changes often occurring in this reach. Three hydrometric stations have been set up in the braided reach, including Huayuankou, Jiahetan and Gaocun, which can be, respectively, abbreviated to HYK, JHT and GC. Based

on the difference in geographical positions and boundary conditions, the braided reach is commonly divided into three sub-reaches, including the upper sub-reach from MJ to HYK, the middle sub-reach from HYK to JHT, and the lower sub-reach from JHT to GC (Figure 1b). In general, the braided reach is characterised by a wide and shallow cross-sectional geometry, a frequently shifting main-channel, and a rapid changing channel planform [19,33].

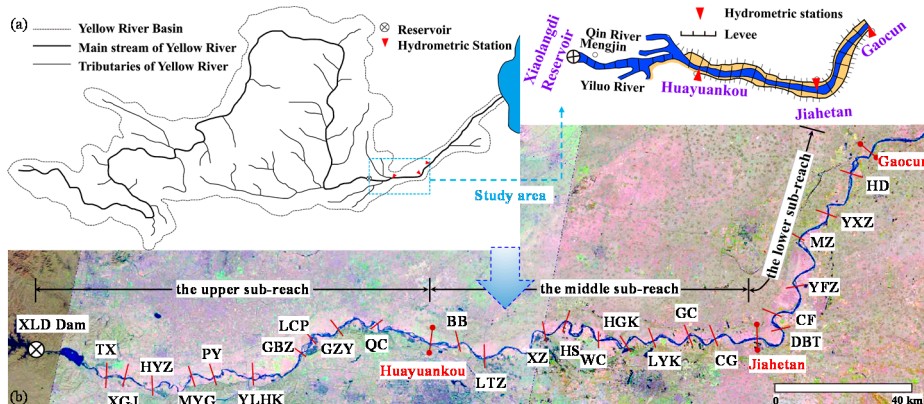

**Figure 1.** Maps of the Lower Yellow River: (**a**) Planview of the YR; and (**b**) Remote sensing image showing the braided reach in the LYR.

In the late 1990s, a large-scale comprehensive water conservancy project, the Xiaolangdi Reservoir, was constructed, which also has the other functions of water supply, irrigation and power generation, in order to control extreme floods and reduce the sedimentation rate in the LYR. Due to a large quantity of sediment trapped in the Xiaolangdi Reservoir since October 1999, the flows with low sediment concentrations were released into the LYR. The LYR therefore underwent continuous channel degradation [10,11]. Furthermore, the degree of channel migration adjusted due to the varying regime of water discharge and sediment concentration.

Flow intermittence refers to the phenomenon of water shortages at some sections of a large river during certain time periods, with the specific performance of low water discharge and dry river bed. Precipitation has been low in most areas of the Yellow River basin since the 1970s, and the situation has deteriorated even further since 1990. Since there is a substantial increase in agricultural irrigation water, the downstream runoff has dropped sharply, which has caused the occurrence of flow intermittence with extremely low discharges in the LYR. Relevant data show that since the 1960s, the annual runoff of the LYR into the sea has gradually decreased from 57.5 billion $m^3$ in the 1960s to 18.7 billion $m^3$ in the 1990s, with a reduction of 67%. Frequent seasonal flow intermittence in the LYR began in the early 1970s, especially in the late 1990s. In 1997, the period of flow intermittence reached 226 days at the section of Lijin in the LYR, and the phenomenon of flow intermittence occurred at the section of JHT, with the period of very low discharges lasting 18 days. After 1999, the phenomenon of flow intermittence did not occur in the LYR, owing to various measures such as the operation of the Xiaolangdi Reservoir, and the establishment of water regulation rules.

## 3. Materials and Methods

### 3.1. Data Sources

#### 3.1.1. Hydrological Data

Hydrological data at three hydrometric stations over the period 1986–2016 were collected, covering daily mean discharges and sediment concentrations, and flood-season and hydrological-year average discharges and sediment concentrations were calculated. In general, the regime of flow and sediment entering the braided reach can be described

approximately by the water discharge and sediment concentration at the Huayuankou station (Figure 2).

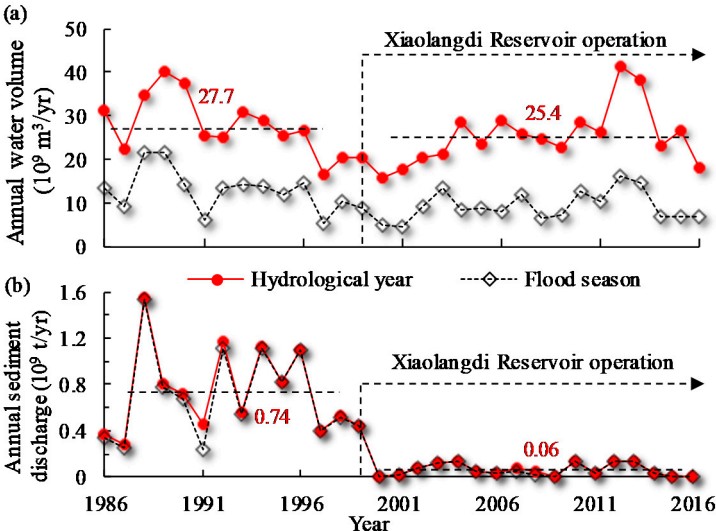

**Figure 2.** Variation in the condition of flow and sediment at HYK: (**a**) water volume; (**b**) sediment discharge.

During the reservoir operation period (1999–2016), the annual mean water volume at HYK was $25.4 \times 10^9$ m$^3$/yr, with the mean value in flood seasons comprising 36%, which was only 6.5% lower than the value of $27.7 \times 10^9$ m$^3$/yr in 1986–1999. However, the sediment amount entering the LYR decreased dramatically because a large quantity of sediment was detained by the Xiaolangdi Reservoir, with the accumulated deposition volume of $3.90 \times 10^9$ t. The average annual sediment discharge at HYK declined from $0.74 \times 10^9$ t/yr (1986–1999) to $0.06 \times 10^9$ t/yr (1999–2016), with the mean value in flood seasons accounting, respectively, for 96% and 95% during these two periods. Before the Xiaolangdi Reservoir operation, the variation in water volume was not totally consistent with the variation in sediment discharge. For example, the annual water volume in 1988 ($34.9 \times 10^9$ m$^3$) was smaller than that in 1989 ($40.3 \times 10^9$ m$^3$), and contrarily the annual sediment discharge in 1988 was significantly greater than that in 1989. After the reservoir operation, the annual water volume entering the LYR changed slightly with a reduction of 8%, whereas there was a huge reduction of 92% in annual sediment discharge. Furthermore, it should be noted that the non-equilibrium suspended load transport had an overwhelming domination over the bed load, mainly because the mean ratio of suspended load to total load was greater than 99.5%, according to the observed data presented in the previous analysis [36]. Therefore, the effect of bed load transport was neglected in the following analysis.

### 3.1.2. Cross-Sectional Topography

Profiles at 28 sedimentation sections measured after the flood seasons along the braided reach were obtained, as well as the amounts of cumulative bed deformation volume in the braided reach and the whole LYR from 1986 to 2016.

In order to better monitor channel deformation, sedimentation sections have been set up along the LYR, with the topographic surveys at cross-sections being conducted prior and post to each flood season [10]. As shown in Figure 3a, the mean bed elevation at the Liuyuankou section of the braided reach was risen, caused by the continuous channel aggradation during the period 1986–1999, with the main-channel shrinking remarkably and the corresponding mean bed level elevated by 1.28 m. However, both bank erosion and bed incision occurred at Liuyuankou during the period 1999–2016, with the corresponding bankfull depth increasing from 1.40 to 3.45 m because of the recent continuous bed incision

(Figure 3b). The number of sedimentation sections displayed a gradually increasing trend after 1999, which greatly improved the monitor accuracy of channel deformation. Before the implementation of the Xiaolangdi Reservoir, there were initially 28 sedimentation sections, with an average distance between two consecutive sections of about 10 km. After the reservoir operation in October 1999, the sedimentation-section number increased gradually, and it was finally equal to 145 after the year of 2005, with a mean spacing of approximately 2 km. Using these cross-sectional profiles surveyed repeatedly, it is easy to obtain the volume of bed deformation over a specified period.

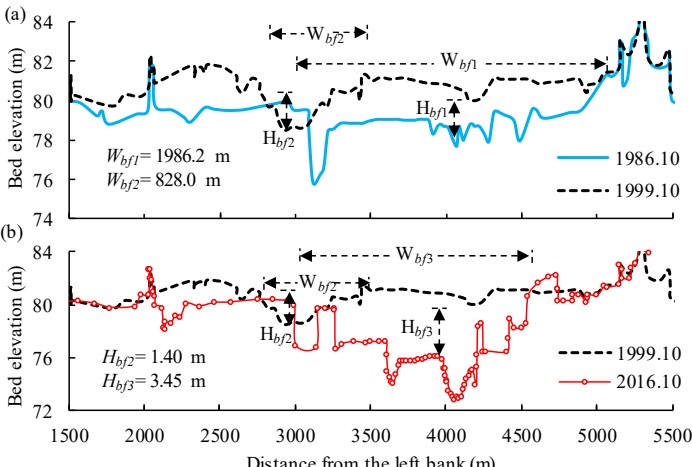

**Figure 3.** Temporal variations in the cross-sectional profile at the Liuyuankou section: (**a**) during the period 1986–1999; and (**b**) during the period 1999–2016.

The braided reach in the LYR underwent two contrasting stages of bed deposition and erosion in 1986–2016, leading to significant variations in lateral shifting of bankline, thalweg, main streamline and channel centerline. The continuous process of channel aggradation in 1986–1999 was caused by the unfavourable matching relationship between water discharge and the corresponding sediment concentration, and the cumulative volume of sediment deposition was $1.51 \times 10^9$ m$^3$ during this period, which comprised 71.9% of the total value in the LYR (Figure 4). However, significant channel degradation occurred in this reach after the onset of the reservoir in 1999, with the cumulative channel scour volume of $1.38 \times 10^9$ m$^3$ comprising 72.2% of the total value in the LYR. The flood-season cumulative volume of channel scour was $0.65 \times 10^9$ m$^3$, which was approximately equivalent to the value of $0.72 \times 10^9$ m$^3$ in non-flood seasons. Since the degree of channel deformation in flood seasons was of equal importance as that in non-flood seasons, the hydrological-year average parameters of flow and sediment over the period 1986–2016 were used in the following analysis.

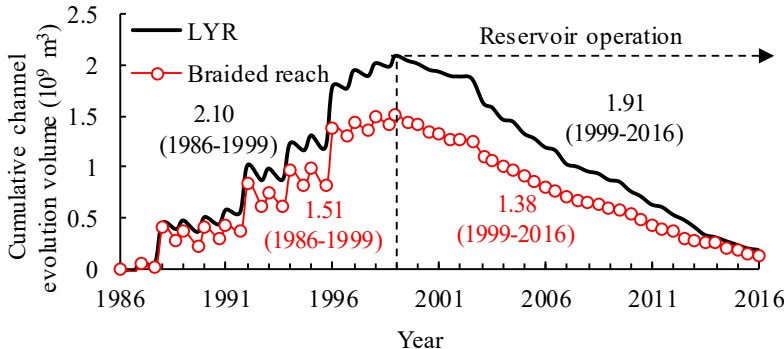

**Figure 4.** Cumulative volume of bed deformation in the braided reach and the LYR during the period 1986–2016.

### 3.1.3. Remote Sensing Images

Remote sensing images are characterized by real-time, accurate, and intuitive information, which are often used to recognize land-water boundaries to depict the process of channel migration [21]. Combined with the GIS technique, remote sensing images in different periods can be compared, with the characteristics of channel migration being determined for a long time series, which can greatly improve the calculation accuracy of channel migration. Therefore, Thematic Mapper (TM), Enhanced Thematic Mapper (ETM+) and Operational Land Imager (OLI) Landsat scenes with a spatial resolution of 30 m over the period from 1986 to 2016 were collected in this study. Figure 5 shows the local satellite images near the sections of HYK, JHT and GC at different times. These satellite images are archived by the EROS Center of USGS (https://glovis.usgs.gov/) (accessed on 20 June 2020).

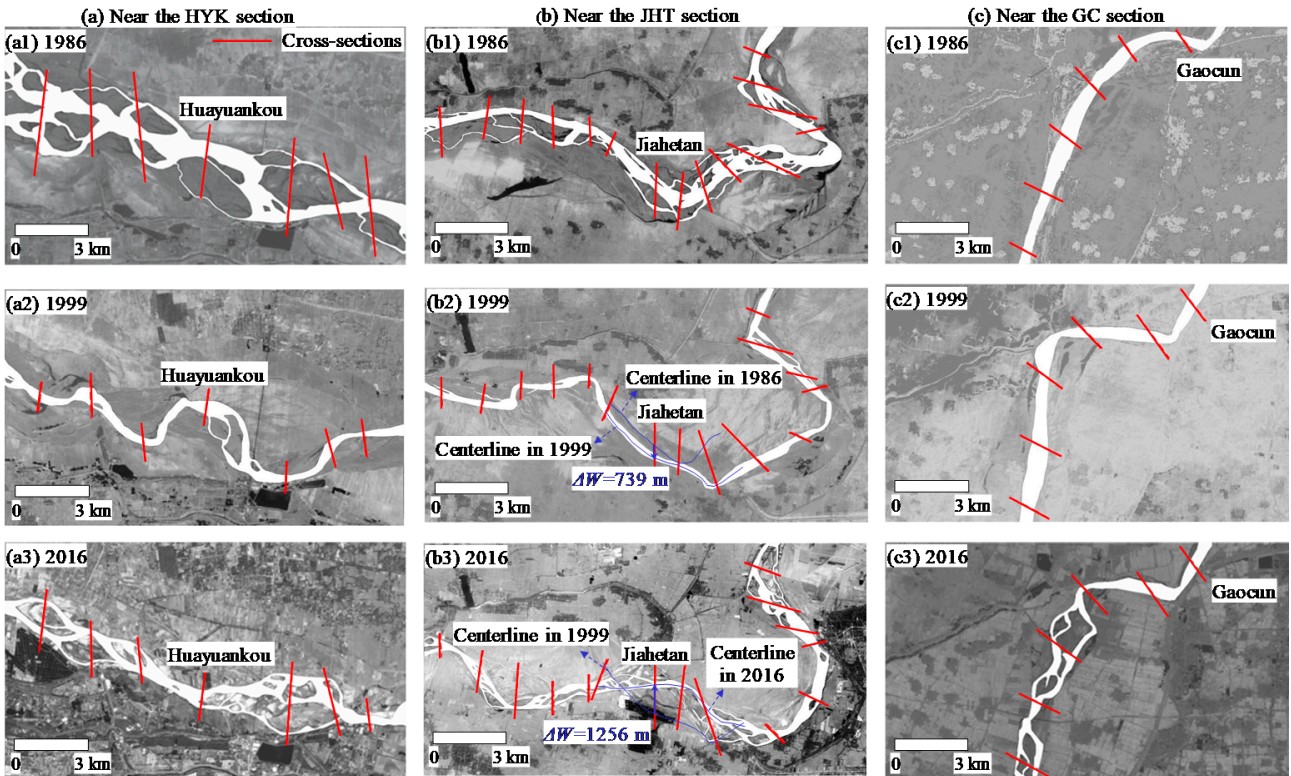

**Figure 5.** Remote sensing images in different years near the sections of: (**a**) HYK; (**b**) JHT; and (**c**) GC.

Since there are many remote sensing images in a hydrological year for a specified satellite sensor, it is necessary to select the most representative images. It is difficult to distinguish the main-channel and floodplain zones during flood seasons owing to large water discharges and high-water levels. However, the post-flood main-channel zone can be detected easily from the remote sensing images, based on the fact that the flow returns to the main-channel zone due to the decreased water discharges. Therefore, the post-flood remote sensing images were selected to extract the banklines of the braided reach. Due to a large area of the study reach, one remote sensing image could not cover the whole study area, and two satellite images were combined to study the characteristics of channel migration in the braided reach. The detailed information of the selected satellite images is shown in Table 1.

**Table 1.** (**a**) Characteristics of the selected satellite images (Path Row is 123). (**b**). Characteristics of the selected satellite images (Path Row is 124).

| a | | | | | | | |
|---|---|---|---|---|---|---|---|
| No | Acquisition Date | Water Level at HYK (m) | Water Level at JHT (m) | Water Level at GC (m) | Image Type | Spatial Resolution (m) | Path Row |
| 1 | 19861203 | 91.54 | 72.96 | 60.54 | Landsat5-TM | 30 | 123 |
| 2 | 19871222 | 91.45 | 73.03 | 60.72 | Landsat5-TM | 30 | 123 |
| 3 | 19881122 | 92.05 | 73.1 | 60.62 | Landsat5-TM | 30 | 123 |
| 4 | 19891125 | 92.32 | 73.47 | 61.03 | Landsat5-TM | 30 | 123 |
| 5 | 19900824 | 92.41 | 73.73 | 61.48 | Landsat5-TM | 30 | 123 |
| 6 | 19911030 | 92.37 | 73.05 | 60.28 | Landsat5-TM | 30 | 123 |
| 7 | 19921016 | 92.64 | 73.8 | 61.45 | Landsat5-TM | 30 | 123 |
| 8 | 19931019 | 92.66 | 74.04 | 61.43 | Landsat5-TM | 30 | 123 |
| 9 | 19941022 | 92.72 | 75.92 | 61.1 | Landsat5-TM | 30 | 123 |
| 10 | 19951110 | 92.84 | 75.83 | 61.45 | Landsat5-TM | 30 | 123 |
| 11 | 19961027 | 92.47 | 75.83 | 61.25 | Landsat5-TM | 30 | 123 |
| 12 | 19971030 | 92.27 | 75.22 | 60.59 | Landsat5-TM | 30 | 123 |
| 13 | 19981017 | 92.69 | 75.19 | 61.7 | Landsat5-TM | 30 | 123 |
| 14 | 19991129 | 92.42 | 75.78 | 62.15 | Landsat7-ETM+ | 30 | 123 |
| 15 | 20001030 | 92.64 | 76.2 | 62.56 | Landsat7-ETM+ | 30 | 123 |
| 16 | 20011017 | 92.46 | 76.04 | 62.27 | Landsat7-ETM+ | 30 | 123 |
| 17 | 20021004 | 92.02 | 76.17 | 62.12 | Landsat7-ETM+ | 30 | 123 |
| 18 | 20031226 | 91.57 | 75.08 | 61.58 | Landsat7-ETM+ | 30 | 123 |
| 19 | 20041025 | 90.98 | 74.62 | 60.69 | Landsat7-ETM+ | 30 | 123 |
| 20 | 20051215 | 91.04 | 75.05 | 60.74 | Landsat7-ETM+ | 30 | 123 |
| 21 | 20061015 | 91.20 | 74.41 | 60.35 | Landsat7-ETM+ | 30 | 123 |
| 22 | 20071103 | 91.04 | 74.39 | 60.75 | Landsat7-ETM+ | 30 | 123 |
| 23 | 20081121 | 90.61 | 73.89 | 59.92 | Landsat7-ETM+ | 30 | 123 |
| 24 | 20091023 | 90.73 | 73.63 | 60 | Landsat7-ETM+ | 30 | 123 |
| 25 | 20101229 | 90.48 | 73.69 | 59.7 | Landsat7-ETM+ | 30 | 123 |
| 26 | 20111114 | 90.92 | 73.96 | 60.29 | Landsat7-ETM+ | 30 | 123 |
| 27 | 20121031 | 89.96 | 73.25 | 59.68 | Landsat7-ETM+ | 30 | 123 |
| 28 | 20131010 | 90.12 | 73.36 | 59.62 | Landsat8-OLI | 30 | 123 |
| 29 | 20141114 | 89.58 | 72.89 | 59.48 | Landsat8-OLI | 30 | 123 |
| 30 | 20151016 | 88.87 | 72.34 | 58.92 | Landsat8-OLI | 30 | 123 |
| 31 | 20161103 | 89.19 | 72.86 | 59.11 | Landsat8-OLI | 30 | 123 |

| b | | | | | | | |
|---|---|---|---|---|---|---|---|
| No | Acquisition Date | Water Level at HYK (m) | Water Level at JHT (m) | Water Level at GC (m) | Image Type | Spatial Resolution (m) | Path Row |
| 1 | 19861110 | 91.44 | 72.69 | 60.39 | Landsat5-TM | 30 | 124 |
| 2 | 19871113 | 91.6 | 73.15 | 60.91 | Landsat5-TM | 30 | 124 |
| 3 | 19881115 | 92.04 | 73.2 | 60.61 | Landsat5-TM | 30 | 124 |
| 4 | 19891204 | 92.34 | 73.75 | 61.35 | Landsat5-TM | 30 | 124 |
| 5 | 19901121 | 92.36 | 73.76 | 61.49 | Landsat5-TM | 30 | 124 |
| 6 | 19911007 | 92.34 | 73.3 | 60.94 | Landsat5-TM | 30 | 124 |
| 7 | 19921025 | 92.46 | 73.46 | 61.16 | Landsat5-TM | 30 | 124 |
| 8 | 19931231 | 92.47 | 73.5 | 61.2 | Landsat5-TM | 30 | 124 |
| 9 | 19941031 | 92.83 | 75.87 | 61.16 | Landsat5-TM | 30 | 124 |
| 10 | 19951103 | 92.74 | 75.78 | 61.36 | Landsat5-TM | 30 | 124 |
| 11 | 19961004 | 92.6 | 76.01 | 61.43 | Landsat5-TM | 30 | 124 |
| 12 | 19971007 | 92.62 | 75.77 | 61.71 | Landsat5-TM | 30 | 124 |
| 13 | 19981127 | 92.59 | 75.2 | 61.71 | Landsat5-TM | 30 | 124 |
| 14 | 19991122 | 92.46 | 75.89 | 62.22 | Landsat7-ETM+ | 30 | 124 |
| 15 | 20001226 | 92.5 | 76.12 | 62.49 | Landsat7-ETM+ | 30 | 124 |
| 16 | 20011111 | 91.87 | 75.36 | 61.88 | Landsat7-ETM+ | 30 | 124 |
| 17 | 20021013 | 92.01 | 76.07 | 62.03 | Landsat7-ETM+ | 30 | 124 |
| 18 | 20031101 | 92.36 | 75.65 | 61.69 | Landsat7-ETM+ | 30 | 124 |
| 19 | 20041002 | 90.91 | 74.83 | 60.87 | Landsat7-ETM+ | 30 | 124 |
| 20 | 20051208 | 91.15 | 75.06 | 60.77 | Landsat7-ETM+ | 30 | 124 |
| 21 | 20061008 | 91.29 | 74.44 | 60.48 | Landsat7-ETM+ | 30 | 124 |
| 22 | 20071128 | 91.15 | 74.18 | 60.43 | Landsat7-ETM+ | 30 | 124 |
| 23 | 20081130 | 90.5 | 73.82 | 59.9 | Landsat7-ETM+ | 30 | 124 |
| 24 | 20091016 | 90.66 | 73.82 | 60.06 | Landsat7-ETM+ | 30 | 124 |
| 25 | 20101104 | 90.51 | 73.77 | 59.84 | Landsat7-ETM+ | 30 | 124 |
| 26 | 20111006 | 91.24 | 74.18 | 60.61 | Landsat7-ETM+ | 30 | 124 |
| 27 | 20121008 | 90.59 | 73.77 | 60.22 | Landsat7-ETM+ | 30 | 124 |
| 28 | 20131019 | 90.48 | 73.77 | 59.97 | Landsat8-OLI | 30 | 124 |
| 29 | 20141006 | 89.63 | 73.05 | 59.52 | Landsat8-OLI | 30 | 124 |
| 30 | 20151009 | 88.9 | 72.42 | 58.85 | Landsat8-OLI | 30 | 124 |
| 31 | 20161214 | 88.84 | 72.2 | 58.77 | Landsat8-OLI | 30 | 124 |

### 3.2. Methods

A detailed calculation procedure is proposed herein to quantify migration rates and intensities of channel centerline at section- and reach-scales, using a fusion approach of data from multiple sources at various scales, obtained from the measurements of hydrological data, cross-sectional topography and remote sensing images. This proposed procedure is illustrated in Figure 6, which covers the calculations of characteristic parameters representing the flow-sediment condition, bankfull channel dimensions (width and depth) at reach-scale, section-scale channel migration rate, and reach-scale channel migration rate and intensity.

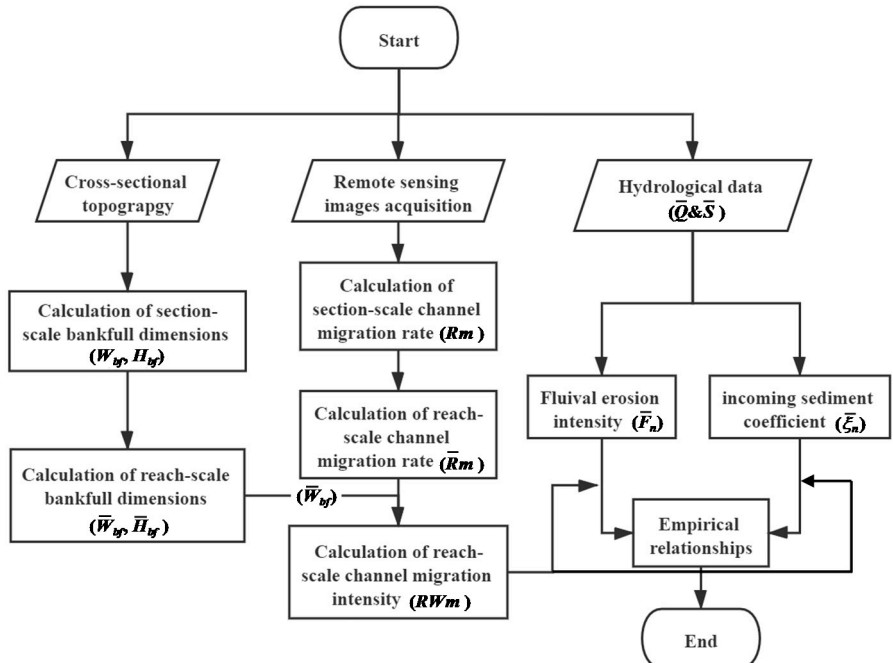

**Figure 6.** Flow chart of calculating the migration rate and intensity of channel centerline.

#### 3.2.1. Hydrological Data

Previous studies show that channel evolution is closely associated with the altered regimes of flow and sediment in alluvial rivers [10,11,37]. The incoming flow-sediment condition in a study reach is generally expressed by a specified function of the mean water discharge and the corresponding sediment concentration. In this study, a characteristic parameter termed as fluvial erosion intensity ($\overline{F}_i$) is used to depict the incoming condition of flow and sediment, which can be written in this form:

$$\overline{F}_i = \left( \overline{Q}_i^{\,2} / \overline{S}_i \right) / 10^4 \tag{1}$$

where $\overline{Q}_i$ is the annual mean water discharge over the $i$th hydrological year; and $\overline{S}_i$ is the corresponding concentration of suspended load.

It is known to us that sediment transport is mainly composed of suspended load instead of bed load in an alluvial river such as the LYR. The relationship between sediment discharge ($Q_s$) and water discharge ($Q$) has been developed in previous studies. Such a relation is usually defined as sediment rating curve (SRC). It commonly takes the form of a power function, and the general relationship between them is expressed by $Q_s = aQ^b$ [38–43]. Syvitski et al. [39] have stated that the SRC is the statistical relationship between suspended sediment discharge and water discharge and, therefore, the SRC is not only for sediment that does not participate in morphological processes. Previous studies also regarded the parameter $a$ as an index of erosion severity in a river channel [44,45],

and the exponent *b* is used to depict the erosion power of a river, both of which reveal that the SRC is related to morphological processes.

In the study, the hydrological data measured under a quasi-equilibrium state in the LYR were collected to calibrate the parameters of the SRC. In this state, the channel geometry generally keeps unchanged and the value of sediment concentration can be approximately equivalent to the magnitude of sediment transport capacity. The calibrated exponent *b* in SRC was roughly equal to 2.0 under a quasi-equilibrium state at sections of HYK and JHT in the LYR ($Q_s = aQ^2$ in Figure 7), with the determination coefficient ($R^2$) between them of 0.66 and 0.70, respectively. It can be seen that the correlations are not very high because the sediment transport capacity is also affected by other factors, but the water discharge is a dominant factor. Therefore, the sediment transport capacity can be written as $S^* \approx Q_s/Q = aQ^1$, and then the base of the fluvial erosion intensity is in the form of $S^*/S = aQ/S$. In order to account for the effect of water discharge on channel deformation, the fluvial erosion intensity is defined by multiplying the water discharge (*Q*) on this basis, that is, $F_i = Q_i^2/S$, with the coefficient being reduced greatly after the reservoir operation, which can be determined in the subsequent relation fitting. This parameter of fluvial erosion intensity has been widely used in previous studies on channel adjustments in the LYR and the Middle Yangtze River [46–48].

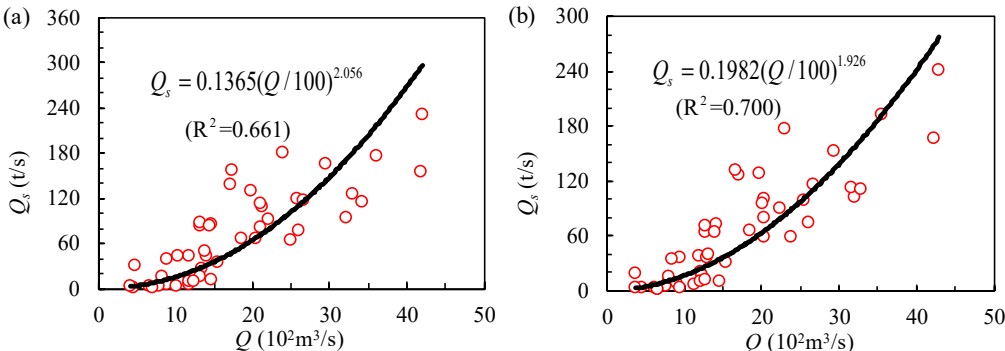

**Figure 7.** Sediment rating curves at sections of: (**a**) HYK; (**b**) JHT.

There is a relaxation or recovery time for the channel to adjust itself under the altered regime of flow and sediment, which is called delayed response [37]. Therefore, channel migration occurs, owing to an accumulative consequence of the earlier conditions of water discharge and sediment concentration, and the mean fluvial erosion intensity ($\overline{F}_n$) during the previous n-year is written as the following form:

$$\overline{F}_n = \frac{1}{n}\sum_{i=1}^{n}\overline{F}_i \tag{2}$$

Previous studies show that channel morphology adjusts along with the variation in the incoming sediment coefficient. Thus, the mean incoming sediment coefficient ($\overline{\xi}_n$) during the previous n-year can also be used to represent the cumulative condition of flow and sediment, which can be written as:

$$\overline{\xi}_n = \frac{1}{n}\sum_{i=1}^{n}\overline{\xi}_i \tag{3}$$

where the incoming sediment coefficient ($\overline{\xi}_i$) refers to the ratio of $\overline{S}_i$ to $\overline{Q}_i$ during the *i*th hydrological year.

### 3.2.2. Calculation of Reach-Scale Bankfull Channel Dimensions

Owing to the fact that remote sensing images cannot quantify the accurate banklines at bankfull level, 28 post-flood profiles of sedimentation cross-sections in 1986–2016 were

adopted to calculate the section-scale bankfull channel widths. Xia et al. [10,11] proposed a suitable identification method to determine the bankfull channel widths at such sections with complex geometry. Furthermore, an integrated approach was utilized in this study to determine the reach-scale bankfull channel width ($\overline{W}_{bf}$), which combines the geometric average based on the log transformation and the weighted mean based on the distance between two consecutive sections [10]. The specific representation of $\overline{W}_{bf}$ is written in this form:

$$\overline{W}_{bf} = \exp\left( \frac{1}{2L_1} \sum_{i=1}^{N_1-1} \left( \ln W_{bf}^{i+1} + \ln W_{bf}^{i} \right) \Delta x_i \right) \tag{4}$$

where $W_{bf}^i$ is the bankfull channel width at the $i$th section; $\Delta x_i$ is the distance between the $i$th and ($i$ + 1)th sections; and $L_1$ is the total channel length in the study reach. It is assumed that the study reach covered $N_1$ cross-sections, and $N_1 = 28$ was used in this analysis. This method can guarantee the continuity of bankfull channel dimensions, as well as avoid the effect on the calculation of bankfull channel dimensions at reach-scale caused by the varied spacing between two sections. In addition, the bankfull channel depths ($H_{bf}^i$ and $\overline{H}_{bf}$) at section- and reach-scales can also be determined using a similar approach. Using the above calculation approach, Figure 3 indicates the section-scale bankfull channel depths and widths at Liuyuankou in three years.

### 3.2.3. Calculation of Section-Scale Channel Migration Rate

Section-scale bankfull discharges vary significantly along the braided reach owing to the difference in cross-sectional geometry [10,11], and it is difficult to identify accurately the river bankline and the main-channel zone at bankfull level from a specified satellite image. Therefore, channel migration can usually be represented by the migration of channel centerline, which can be determined from the channels at low water levels extracted from the post-flood satellite images.

It should be noted that remote sensing images obtained from the Landsat scenes were all Level 1T terrain-correction images, with topography, systematic radiation and geometric correction being conducted. The following steps were used to process the selected remote sensing images [7,21]. Firstly, the NDWI approach developed by McFeeters [49] was used to identify the water body area; secondly, the banklines are defined as the boundary of the extracted water body, with the water surface being regarded as the river channel zone (Figure 5); thirdly, Software Envi was used to draw the left and right banklines of the braided reach, with the corresponding positions of 145 cross-sections being marked according to their latitude and longitude, and then the extracted banklines and the positions of cross-sections were imported into Software ArcMap to determine their coordinates; finally, two intersection points of $(X_{Lk}, Y_{Lk})$ and $(X_{Rk}, Y_{Rk})$ between two banklines and the $k^{th}$ section were determined using the corresponding Fortran code, with the middle site of these two points being defined as the center of the section $(X_{Ck}, Y_{Ck})$. The connection line for the middle point of each section is defined as the channel centerline (Figure 8). The ratio of the width ($\Delta W_k$: m) between the middle points of a section to the specified period ($\Delta t$: yr) is termed as the section-scale migration rate ($R_m^k$: m/yr), which can be written in this form:

$$R_m^k = \Delta W_k / \Delta t \tag{5}$$

where $\Delta t$ is the time interval between two different dates ($t_1$ and $t_2$) of interest, and $\Delta t = t_2 - t_1$. The middle point at a section is considered to migrate toward the left side if this point is located on the left side after a period of $\Delta t$. For example, the migration width of channel center at JHT was 739 m during the period 1986–1999, with the mean migration rate of 57 m/yr calculated by Equation (5), as shown in Figure 5(b2). The migration width of channel center at JHT was 1256 m in 1999–2016, with the corresponding mean migration rate of 74 m/yr, as shown in Figure 5(b3).

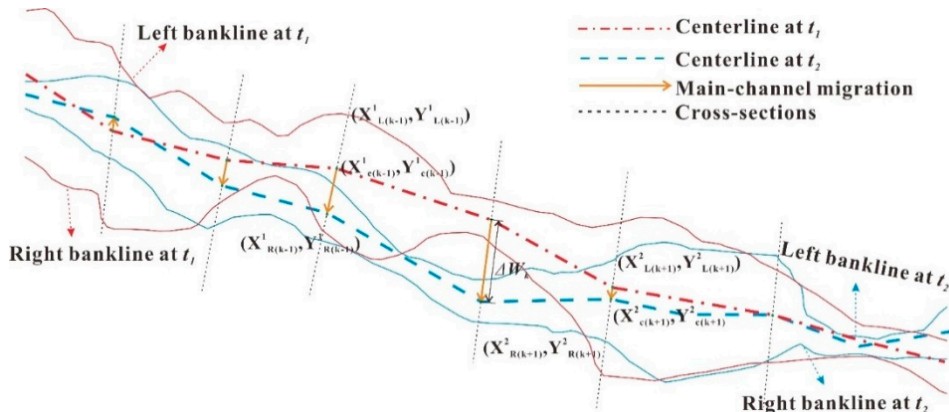

**Figure 8.** Sketch map for determination of the centerline migration rate at section-scale.

3.2.4. Calculation of Reach-Scale Channel Migration rate and Intensity

The reach-scale migration rate of channel centerline ($\overline{R}_m$: m/yr) over the specified period $\Delta t$ can be determined by the similar reach-averaged approach. Since the migration rate of channel centerline at section-scale over a specified period may be zero, the equation for calculating the reach-scale migration rate can be written as:

$$\overline{R}_m = \frac{1}{2L_2} \sum_{i=1}^{N_2-1} \left( R_m^k + R_m^{k+1} \right) \Delta l_k \tag{6}$$

where $N_2$ is the number of the refined cross-sections; $\Delta l_k$ is the distance between the $k$th and $(k + 1)$th sections; $L_2$ is the total length of channel centerline in the study reach determined based on the remote sensing images. It should be pointed out that the calculation accuracy of the reach-scale migration rate depends on the number of sections adopted in the reach, and a larger number of cross-sections can produce a higher calculation accuracy [50]. As mentioned above, the number of the measured sedimentation sections after 2004 was equal to 145 in the braided reach. Therefore, $N_2 = 145$ was used in Equation (6).

It is unreasonable to assess the channel stability only using the variation in migration rate of channel centerline, because the adjustment in the bankfull channel width at reach-scale can influence the rate of centerline migration. Thus, a ratio of the reach-scale migration rate of channel centerline to the mean bankfull channel width at reach-scale in a period of $\Delta t$ can be calculated as a relative migration rate. This ratio can represent the lateral mobility degree of main-channel [2], and it is defined as the reach-scale migration intensity of channel centerline ($RW_m$: m/(yr·m)), which can be written as:

$$RW_m = \frac{\overline{R}_m}{\left( \overline{W}_{bf1} + \overline{W}_{bf2} \right)/2} \tag{7}$$

where $\overline{W}_{bf1}$ and $\overline{W}_{bf2}$ are the reach-scale bankfull widths at dates of $t_1$ and $t_2$.

## 4. Results and Discussion

In this study, the above methods were employed to quantify the section- and reach-scale morphometric parameters, covering the migration direction, rate and intensity of channel centerline in the braided reach in 1986–2016.

*4.1. Migration Direction of Channel Centerline*

Migration directions at the typical sections of HYK, JHT and GC were determined during the period 1986–2016. The total numbers of leftward migration at three sections were 17, 16, and 13, respectively, while the numbers of rightward migration were 13, 14, and 16 (Figure 9a). These results indicate that the probability of the channel centerline

migrating toward the left bank was approximately equal to the probability of rightward migration from a long-term sequence.

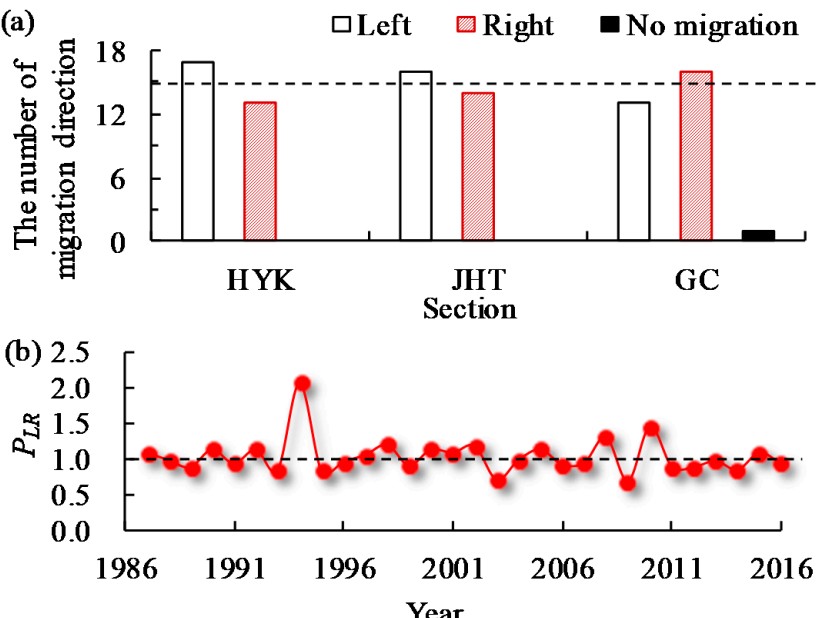

**Figure 9.** Variations in the section- and reach-scale centerline migration direction: (**a**) the number of each migration direction at typical sections; (**b**) reach-scale migration probability (PLR).

Furthermore, the migration directions at 145 sections in the braided reach of the LYR were also determined annually from 1986 to 2016. The number of cross-sections at which the channel centerline migrated toward the left or right bank over a specified period was defined, respectively, as $N_L$ or $N_R$, and the ratio ($P_{LR}$) was calculated by the value of $N_L/N_R$. At these sections, the values of $\overline{N}_L$ and $\overline{N}_R$ were, respectively, 73 CSs/yr and 72 CSs/yr during the period 1986–2016, with the ratio ($P_{LR}$) equaling 1.01 (Figure 9b). Therefore, it can be concluded that the probability of channel migration toward the left or right side was generally equivalent at both section- and reach-scales. It is mainly attributed to this fact that a number of central bars were widely distributed and the riverbed consisted of fine sand, which caused remarkable channel adjustments in the braided reach [3]. This finding reveals that the migration direction of channel centerline in the braided reach was reciprocating, which fully reflected the self-adjustment characteristics of this reach [1].

*4.2. Migration Rate of Channel Centerline*

In 1986–1999, the braided reach underwent a remarkable aggradation process, which led to the centerline of the channel shifting significantly. In 1999–2016, there was a significant channel degradation process in this reach, with the degree of channel migration decreasing gradually [51]. Figure 10a shows the migration rates of channel centerline at the sections of HYK, JHT and GC during the period from 1986 to 2016. These results indicate that: (i) the migration rates of channel centerline were a little higher in the years of 1988, 1992 and 1996 due to the occurrence of hyperconcentrated flood events before the reservoir operation, and the corresponding rates at three sections were 973, 190 and 50 m/yr in 1988, 1224, 1284 and 6 m/yr in 1992, and 1011, 49 and 51 m/yr in 1996, respectively, and the average migration rates of channel centerline in 1986–1999 were 540, 341 and 92 m/yr; (ii) after the Xiaolangdi Reservoir operation (1999–2016), the maximum migration rate at HYK reached 1049 m/yr in 2000 owing to the recession of a sharply curved bend near the HYK section; similarly, the migration rate at JHT increased gradually from 1999 to 2002, with the maximum value approximating 961 m/yr in 2002, and the migration rate decreased gradually after a sharply curved bend was regenerated; the average annual migration rates of channel centerline at HYK, JHT and GC were 244, 296 and 48 m/yr over

the period 1999–2016, with the corresponding reduction degrees of 55%, 13% and 48%; and (iii) moreover, the migration rates at the sections of HYK and JHT were much larger than the values at the section of GC during the period 1986–2016. The GC section had a narrower and deeper geometry and the local river training works reduced the degree of channel migration, as compared with the cross-sections of HYK and JHT without the protection of effective river training engineering.

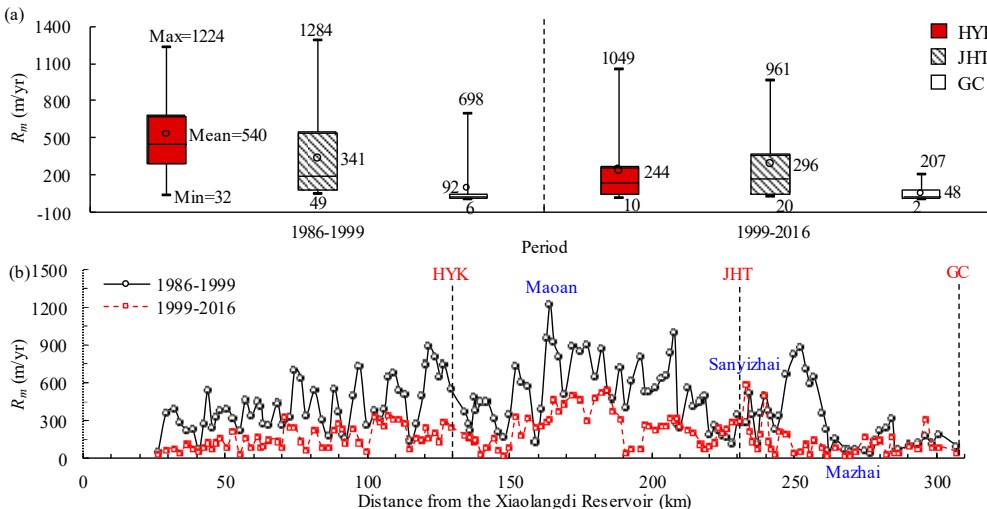

**Figure 10.** Temporal variations in the section-scale migration rate of channel centerline: (**a**) at sections of HYK, JHT and GC; (**b**) at all the sections in the braided reach.

In general, the migration rates of channel centerline varied greatly along the braided reach, and were influenced by various factors such as the conditions of water discharge and sediment concentration, and channel boundary conditions [3]. During the period 1986–1999, the largest migration rate of channel centerline occurred at the Maoan section situated 66.2 km upstream of JHT, with the value of 1225 m/yr. After the Xiaolangdi Reservoir operation, the maximum migration rate (exceeding 590 m/yr) in 1999–2016 occurred at the Sanyizhai section located 2.9 km downstream of JHT, while the minimum value was about 20 m/yr occurring at the Mazhai section located 37.6 km downstream of JHT. In addition, only 19 of the 145 cross-sections after the reservoir operation had a larger migration rate than the value before the reservoir operation, which indicates that the migration degree of channel centerline was reduced remarkably, with the channel braiding property weakening, caused by the water impoundment and sediment retention of the Xiaolangdi Reservoir. Figure 10b also reveals that the rate of channel migration was the most dramatic in the HYK–JHT sub-reach. One of the most important reasons can be explained as follows. The critical shear stress of the river bank soil is generally much lower than the near-bank shear stress of the flow because of its weak erosion-resistance and lower clay content, and the shear strength of bank soil can be determined by the internal friction angle and cohesion, with their values decreasing greatly with an increase in water content [52]. Therefore, the riverbanks in this sub-reach were prone to being scoured and collapsing, which made it possible for the channel centerline to shift from one branch to another. Figure 10 indicates that the characteristics of channel migration at a specified section or in a local sub-reach made it difficult to represent the migration trend of a whole study reach, and that the reach-scale migration rate needs to be investigated.

Subsequently, the reach-scale migration rates of channel centerline in the braided reach from 1986 to 2016 were calculated using Equation (6), as shown in Figure 11a. During the continuous channel aggradation period 1986–1999, the largest reach-scale channel migration rate was 659 m/yr, which occurred in 1988 mainly due to the occurrence of a hyperconcentrated flood event, and the average annual channel migration rate was 410 m/yr in 1986–1999. After the reservoir operation, the maximum reach-scale migration

rate (303 m/yr) occurred in 2003, and it was mainly related to strong flood processes. Five flood events with low sediment concentrations and large water discharges occurred in the 2003 flood season, which caused intensive scour and migration processes. The average annual migration rate at reach-scale was 185 m/yr in 1999–2016, with a reduction of 55%, as compared with the pre-dam value in 1986–1999.

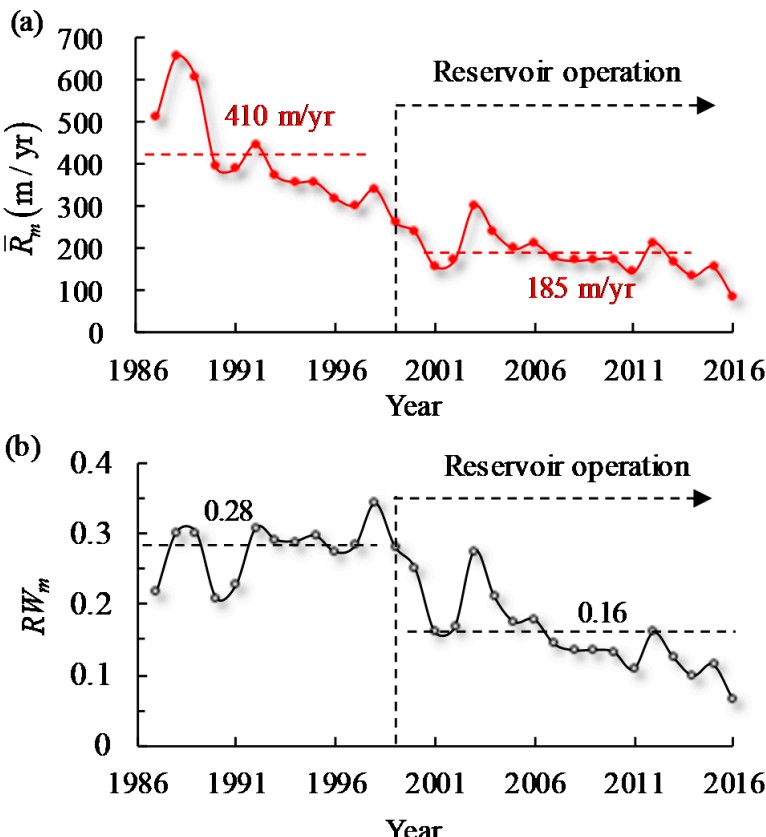

**Figure 11.** Temporal variations in the reach-scale channel migration parameters of: (**a**) rate; (**b**) intensity.

In order to further explore the temporal and spatial characteristics of channel migration in the braided reach, the average migration rates of three sub-reaches were calculated using the above-mentioned reach-averaged method, as shown in Figure 12a. In terms of temporal variation, the results show that the migration rate of each sub-reach was relatively large before the reservoir operation, and it tended to have a gradually decreasing trend thereafter. It was mainly related to a significant variation in the flow and sediment regime after the reservoir operation. In terms of spatial variation, the mean migration rates in three sub-reaches were 403, 522 and 274 m/yr before the reservoir operation; over the period 1999–2016, the mean migration rate was the most significant in the HYK–JHT sub-reach ($\overline{R}_m$ = 252 m/yr), while the mean rates in the MJ–HYK and JHT–GC sub-reaches were 160 and 129 m/yr, respectively. On the whole, statistical results of migration rates in these three sub-reaches were consistent with the calculated results at section-scale. Therefore, the migration rate of channel centerline in the braided reach was characterized spatially by the fact that the values in the middle sub-reach were generally larger than those in the upper and lower sub-reaches.

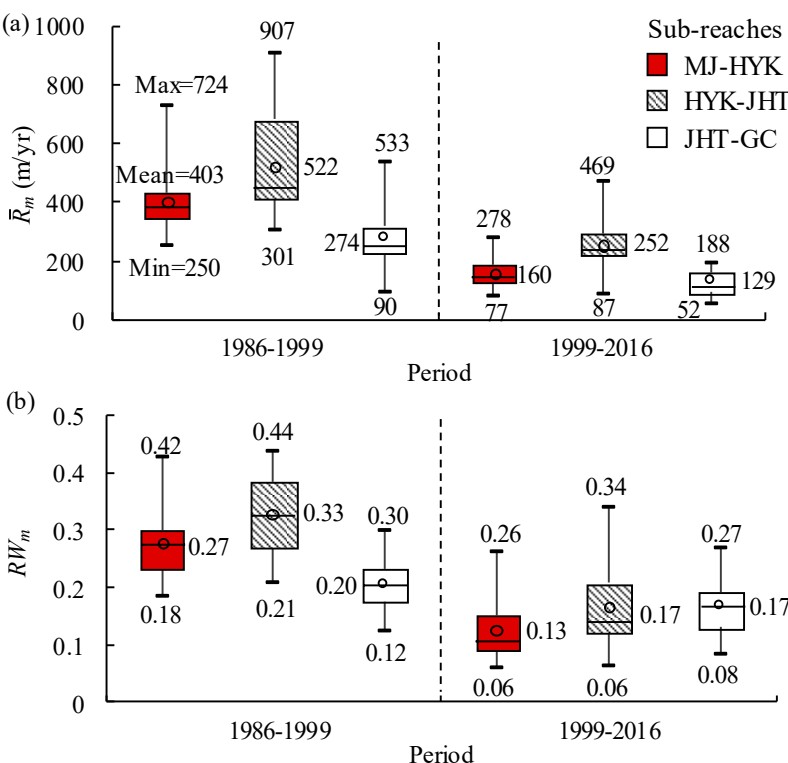

**Figure 12.** Temporal variations in the reach-scale migration parameters of channel centerline in three sub-reaches: (**a**) rate; (**b**) intensity.

### 4.3. Migration Intensity of Channel Centerline

The reach-scale migration intensities of channel centerline over the period 1986–2016 were also calculated using Equation (7). It can be seen in Figure 11b that the variation trend in migration intensity was consistent with the migration rate in 1986–2016. The maximum reach-scale migration intensity of channel centerline was 0.34 m/(yr·m) in 1998, which was higher than the average annual migration intensity of 0.28 m/(yr·m) during the period 1986–1999. After the reservoir operation, the maximum migration intensity reached 0.28 m/(yr·m), occurring in 2003; subsequently, the migration intensity gradually decreased, ranging between 0.07 and 0.21 m/(yr·m). In 1999–2016, the average annual migration intensity of channel centerline was 0.16 m/(yr·m), with a corresponding reduction of 43%, as compared with the average annual value in 1986–1999. This variation trend was mainly related to the reduced sediment concentration and the subsequent channel incision.

In addition, the average annual channel migration intensities in 1986–1999 were, respectively, 0.27, 0.33 and 0.20 m/(yr·m) in the sub-reaches of MJ–HYK, HYK–JHT, and JHT–GC, as shown in Figure 12b. In 1999–2016, the average annual migration intensities in three sub-reaches were reduced, respectively, to 0.13, 0.17 and 0.17 m/(yr·m), with the corresponding reduction degrees of 52%, 48% and 15%. The JHT–GC sub-reach had the smallest channel migration rate, but the mean migration intensity was relatively large, because the bankfull channel width in this sub-reach was only half of the other two sub-reaches, resulting in a higher channel migration intensity.

### 4.4. Influencing Factors of Channel Migration Intensity

Channel evolution covers various deformation components such as adjustments in planform and cross-sectional geometries, and channel migration is a critical component of channel planform adjustments in alluvial rivers [3,13]. Factors influencing channel migration mainly include the variations in channel boundary and incoming flow-sediment



regime [1,53,54]. These influencing factors are presented herein to investigate the variation characteristics in migration intensity of channel centerline in the braided reach.

### 4.4.1. Effect of Channel Boundary Conditions

Channel boundary conditions generally refer to different geomorphological factors, including bed-material composition, geomorphic coefficient, the level difference between main-channel and floodplain, as well as the longitudinal channel slope [1,3]. In addition, the construction of river training works would also have an impact on the channel evolution of a local reach.

The river training works implemented in the Lower Yellow River are mainly divided into three types, including shoal-protection works, flow guide works and levee protection works [1,19,55]. Hu et al. [55] found that the migration rate of main-channel at HYK reached 7.5 km/yr under natural conditions; however, the migration rate was reduced to 3 km/yr after the implementation of river training works in the early 1990s. Further investigations reveal that only 7 of a total of 110 river regulation projects still had a certain effect on the mainstream migration in the braided reach [13], which indicates that it was impossible to achieve long-term channel stability merely controlled by river training works. It is confirmed that both incoming flow-sediment regime and channel boundary conditions are two main factors to control the channel migration rate in the braided reach, and it is difficult to completely change the inherent adjustment characteristics of rivers through limited artificial projects which only play an auxiliary role [19].

With the purpose of investigating the relationship between migration intensity of channel centerline and composition of bed material, the mean post-flood median diameters of bed material ($D_{50}$) at HYK, JHT and GC were calculated annually as the representative diameter of bed material. Before the reservoir operation, channel aggradation occurred in the downstream reach, with the bed-material composition gradually becoming fine, and then an intensive process of channel degradation occurred, characterized by evident coarsening of the bed-material. The outcomes presented in Figure 13a show that the migration intensity of channel centerline decreased as the median diameter increased, with the determination coefficient between them of $R^2 = 0.66$, which qualitatively reflected the restrictive effect of bed-material composition on channel migration in the braided reach. It is also one of the reasons explaining a decrease in the migration intensity of channel centerline since 1999.

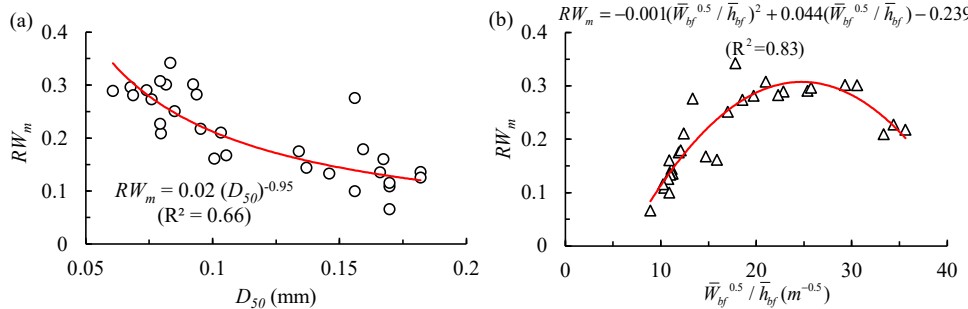

**Figure 13.** Relationships between migration intensity and various channel boundary conditions of: (**a**) median diameters of bed material ($D_{50}$); (**b**) geomorphic coefficient ($\overline{W}_{bf}^{0.5}/\overline{h}_{bf}$).

In addition, geomorphic coefficient ($\sqrt{\overline{W}_{bf}}/\overline{h}_{bf}$) was used to represent the cross-sectional geometry [10]. A polynomial function relationship can be established between $RW_m$ and geomorphic coefficient ($\sqrt{\overline{W}_{bf}}/\overline{h}_{bf}$), as shown in Figure 13b. The migration intensity of channel centerline first generally increased with the increasing geomorphic coefficient, whereas it reached the maximum value when the magnitude of geomorphic coefficient was equal to about 25 m$^{-0.5}$, and then it decreased with the increasing geomorphic coefficient, with $R^2 = 0.83$. It can be seen that the migration degree was stronger in a wider and shallower channel geometry when geomorphic coefficient ranged within a

certain limitation; however, the channel showed a narrower and deeper geometry owing to the recent channel degradation in 1999–2016, which decreased the migration degree of channel centerline.

### 4.4.2. Effect of the Altered Flow and Sediment Regime

After the Xiaolangdi Reservoir operation, the flow and sediment regime entering the braided reach was altered significantly, leading to a great reduction in the amount of sediment discharge at HYK. Consequently, the downstream channel adjusted from the continuous deposition state to the recent degradation state, and the volume of channel scour accounted for more than half of the total volume of the whole LYR. Therefore, the varying regime of flow and sediment was the primary factor causing adjustments in cross-sectional geometry. In the current study, Equations (1)–(3) were used to analyze the effect of the incoming flow and sediment regime on the migration intensity.

In order to study the relationship between the intensity of channel migration and the incoming regime of flow and sediment, these annual mean parameters of flow and sediment at HYK from 1986 to 2016 were calculated, covering the discharge, sediment concentration, incoming sediment coefficient, and fluvial erosion intensity. Various relationships were then presented between these flow-sediment parameters and the reach-scale migration intensity of channel centerline ($RW_m$), which displayed both single and combined effects of flow and sediment conditions on the reach-scale migration intensity of channel centerline, as shown in Figure 14. These relations in Figure 14a,b can be written as:

$$RW_m = 0.92\left(\bar{\bar{\zeta}}_2\right)^{0.34} \tag{8a}$$

$$RW_m = 0.39\left(\bar{F}_2\right)^{-0.31} \tag{8b}$$

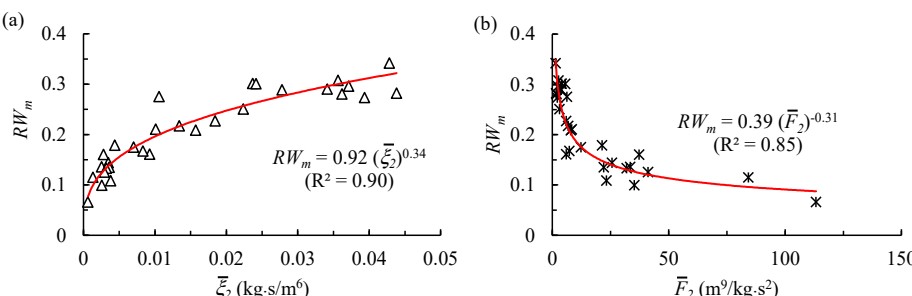

**Figure 14.** Relationships between the channel migration intensity and various incoming flow-sediment factors of: (**a**) the previous two-year average incoming sediment coefficient ($\bar{\bar{\zeta}}_2$); (**b**) the previous two-year average fluvial erosion intensity ($\bar{F}_2$).

It indicates that: the value of $RW_m$ increased with a larger previous two-year mean incoming sediment coefficient ($\bar{\bar{\zeta}}_2$), with $R^2$ = 0.90 (Figure 14a), and decreased as the previous two-year mean fluvial erosion intensity ($\bar{F}_2$) increased, with $R^2$ = 0.85 (Figure 14b). It comes to a conclusion from Figure 14 that the incoming sediment coefficient or fluvial erosion intensity is an important factor that remarkably influenced the migration intensity of channel centerline in the braided reach over the period 1986–2016.

Based on the results calculated using Equation (8a,b), the variation in the migration intensity of channel centerline in the braided reach was reproduced. Figure 15 indicates that the migration intensities of channel centerline at reach-scale, estimated using Equation (8a,b), can generally agree with the measured process in 1986–2016. Moreover, measured migration intensities of channel centerline in 2017 and 2018 were used to verify two empirical relations, as shown in red dots. It shows that the migration intensities of channel centerline at reach-scale, estimated using Equation (8a,b), can generally agree with the measured data in 2017 and 2018, with the relative errors of 1.5% (in 2017) and 21% (in 2018) of Equation (8a), and 20% (in 2017) and 37% (in 2018) of Equation (8b). Therefore, these

two empirical relations can be used to predict the migration intensity of channel centerline in the braided reach.

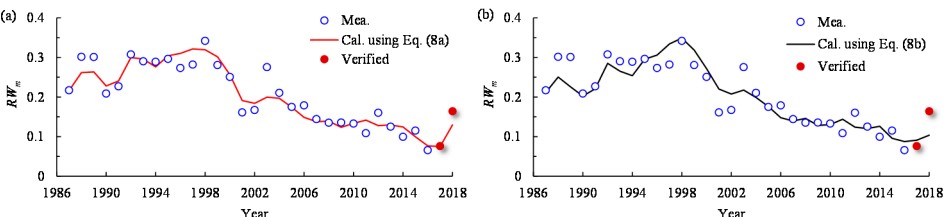

**Figure 15.** Comparisons between the measurements and the reach-scale channel centerline migration intensities calculated using: (**a**) Equation (8a); (**b**) Equation (8b).

## 5. Conclusions

Channel migration plays a vital role in adjustments of channel planform in the braided reach. In this study, a detailed calculation procedure was proposed to determine migration rate and intensity of channel centerline, using remote sensing images and cross-sectional topography. Migration rates and intensities of channel centerline at section- and reach-scales in the braided reach in 1986–2016 were calculated using the proposed procedure. The conclusions are presented as follows:

(i) The probability of channel migration towards the left or right side was almost equivalent over the period 1986–2016, and section-scale migration rates in the middle sub-reach were generally higher than the values in the upper and lower sub-reaches.

(ii) The average annual reach-scale migration rate of channel centerline was 410 m/yr over the period 1986–1999, and it was reduced to 185 m/yr over the period 1999–2016 due to the recent channel scour caused by the implementation of the reservoir. The reach-scale migration intensity of channel centerline underwent a similar adjustment, with the average annual migration intensity decreasing from 0.28 m/(yr·m) in 1986–1999 to 0.16 m/(yr·m) in 1999–2016.

(iii) The incoming flow and sediment regime was a dominant factor influencing the migration intensity of channel centerline, although channel boundary conditions could also influence the migration intensity. The reach-scale migration intensity of channel centerline was written as a function of the previous two-year mean incoming sediment coefficient or fluvial erosion intensity at HYK, and corresponding parameters were calibrated by the measurements in 1986–2016, with the corresponding determination coefficients of 0.90 and 0.85. Furthermore, the proposed relations were also verified against the measurements in 2017 and 2018. Therefore, reach-scale centerline migration intensities calculated using the proposed relations are generally in close agreement with the measurements.

**Author Contributions:** Methodology, J.X. and Y.W.; formal analysis, Y.W.; writing—original draft preparation, J.X.; conceptualization, M.Z.; writing—review and editing, Y.W., S.D., Z.L. and Z.W. All authors have read and agreed to the published version of the manuscript.

**Funding:** This research was funded by the National Natural Science Foundation of China, grant numbers (51725902, 51579186, 51809196) and the Program of the National Key Research and Development Plan, grant number 2017YFC0405501.

**Institutional Review Board Statement:** Not applicable.

**Informed Consent Statement:** Not applicable.

**Data Availability Statement:** Not applicable.

**Acknowledgments:** The study reported herein was supported by the National Natural Science Foundation of China (Grant Nos. 51725902; 51579186; 51809196), and it was also supported partly by the Program of the National Key Research and Development Plan (Grant No. 2017YFC0405501). The constructive suggestions of the anonymous reviewers are gratefully acknowledged.

**Conflicts of Interest:** The authors declare no conflict of interest.

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
