# Peer review of "Variations in Channel Centerline Migration Rate and Intensity of a Braided Reach in the Lower Yellow River"

_remotesensing, doi:10.3390/rs13091680_

Round 1

Reviewer 1 Report

Very interesting paper, I suggest to continue in a more international cooperation group such research activity.

Author Response

Reply to Reviewer #1’s Comments

Question 1: Very interesting paper, I suggest to continue in a more international cooperation group such research activity.

Answer 1: Many Thanks for your comment and suggestion.

Reviewer 2 Report

 This is an interesting and very good paper on channel migration in the lower Yellow River. It provides a good example of quantitative analysis of channel migration in braided rivers based on a number of data sources derived from historical maps, cross-sectional topography, and remote sensing images. There are many minor English problems likely due to the direct translation from Chinese terms that could be improved.

Miscellaneous comments/suggestions:

Line 61-85: All references in this paragraph are on Yellow River. The authors should mention this first. So that the readers do not have to check which river these are about.

Line 130: “…and finally extends eastward to the Bohai Sea…..” Need to mention what climate is in this region? Semi-arid or wet….?

Line 155: “conservancy project entitled the..” Delete ‘entitled’.

Line 158: Change ‘great’ to ‘large’.

Line 161: remove “hydrographs of”.

Line 170: remove “hydrographs”.

Line 171-172: Suggest change to ” …which has caused the occurrence of flow intermittence with extremely low discharges in the LYR.” Also, the term “flow intermittence” is not clearly defined. It is not a common term familiar to river hydraulicians.

Line 205: Remove “transport of”.

Line 306: Remover “often”.

Sec. 3.2.1:  It will be good to mention that the sediment transport is mainly suspended load comparing to bedload in other rivers and discuss the effects on their SRC curves.

Line 328-330: See above comment.

Fig. 13 b: Could a better relationship instead of a simple power function be obtained?

Line 679:  Remove “6. Patents”.
